# The First Records of *Trissolcus japonicus* (Ashmead) and *Trissolcus mitsukurii* (Ashmead) (Hymenoptera, Scelionidae), Alien Egg Parasitoids of *Halyomorpha halys* (Stål) (Hemiptera, Pentatomidae) in Serbia

**DOI:** 10.3390/biology13050316

**Published:** 2024-05-01

**Authors:** Aleksandra Konjević, Luciana Tavella, Francesco Tortorici

**Affiliations:** 1Center of Excellence, Faculty of Agriculture, University of Novi Sad, Trg Dositeja Obradovica 8, 21000 Novi Sad, Serbia; sashak@polj.uns.ac.rs; 2Dipartimento di Scienze Agrarie, Forestali e Alimentari (DISAFA), University of Torino, largo P. Braccini 2, 10095 Grugliasco, Italy; luciana.tavella@unito.it

**Keywords:** brown marmorated stink bug, foreign insect species, first records, parasitoid list

## Abstract

**Simple Summary:**

In Serbia, there is a growing concern over the Brown Marmorated Stink Bug, which damages plants and creates aggregates in urban areas during the winter season. The bug population has increased, causing significant economic damage to crops. A study conducted in 2022 has discovered nine egg parasitoid species, including two foreign species, *Trissolcus japonicus* and *Tr. mitsukurii*, which were found in only one city in Serbia. This is the first time these species have been recorded in Serbia.

**Abstract:**

Serbia has recently begun facing a serious problem with the Brown Marmorated Stink Bug, *Halyomorpha halys* (Stål), which was first recorded in October 2015. This species belongs to the Pentomidae family and is notorious for causing extensive damage to plants. During the winter, it tends to gather in urban areas, such as houses and different man-made facilities, which has raised concerns among producers and citizens. The population of this species has rapidly increased, causing significant economic damage to cultivated plants. However, despite the alarming situation no natural enemies have yet been identified in Serbia. Therefore, research in 2022 was focused on collecting stink bug eggs to investigate the presence of egg parasitoids. The study identified two foreign Hymenoptera species for the European region, *Trissolcus japonicus* (Ashmead) and *Tr. mitsukurii* (Ashmead) (Scelionidae), recorded for the first time in Serbia. Additionally, the list of egg parasitoid species belonging to the Hymenoptera order includes seven local species: *Anastatus bifasciatus* (Geoffroy), from the Eupelmidae family; *Ooencyrtus* sp., from the Encyrtidae family; and *Telenomus turesis* (Walker), *Tr. basalis* (Wollaston), *Tr. belenus* (Walker), *Tr. colemani* (Crawford), and *Tr. semistriatus* (Nees von Esenbeck), from the Scelionidae family. In total, nine egg parasitoid species were, for the first time, reported as parasitizing *H. halys* and related species in Serbia.

## 1. Introduction

From its native range in eastern Asia, *Halyomorpha halys* (Stål) (Hemiptera, Pentatomidae) has spread throughout the Holarctic region and in a few countries in South America [1], to the point that it has become a worldwide invasive pest with more than 100 host plants reported [2].

Due to the severe damage caused by the stink bug and the resulting economic losses, the use of insecticides has increased as the control of this pest relies mainly on pesticides [3,4]. Because of the application of pesticides, including broad-spectrum insecticides, beneficial insects are also killed and integrated pest management programs are negatively impacted [5].

Among the natural enemies of *H. halys*, considerable emphasis has been placed on egg parasitoids. In its native range, the stink bug is regulated by several egg parasitoids. *Trissolcus japonicus* (Ashmead) and *Trissolcus mitsukurii* (Ashmead) are predominant and the most effective parasitoids in the native area [6,7,8].

In Europe, associated parasitoids belonging to Scelionidae (*Trissolcus* spp., *Telenomus* spp.), Eupelmidae (*Anastatus* spp.), and Encyrtidae (*Ooencyrtus* spp.) were found to parasitize *H. halys* eggs [9,10,11,12,13,14,15,16], but they were not fully effective in controlling *H. halys* populations. In particular, *Anastatus bifasciatus* (Geoffroy) and *Trissolcus kozlovi* Rjachovskij are the two native egg parasitoids in Europe that have shown some potential for the control of *H. halys* and have been considered for augmentative biological control strategies [10,13,17,18,19,20].

Several studies have been carried out to explore the ability of parasitoids to attack *H. halys* eggs in their native area as well as in the newly invaded areas worldwide [9,11,13,21]. Actually, only the two Asian species *Tr. japonicus* and *Tr. mitsukurii* appear to be promising candidates as biological control agents [8,22,23,24,25]. Since 2014, *Tr. japonicus* started to be detected in the USA [26], then in 2017 in Europe [18], and now the distribution area is widespread [27,28,29]. The first detection of *Tr. mitsukurii* outside the native area (i.e., in Italy) was in 2016 [30], and then the parasitoid spread to France [31].

In Serbia, *H. halys* was detected in October 2015 [32]. In the following years, research was conducted to assess the distribution of the species and the damage to local crops [33], when it was confirmed that the species had become established in the newly invaded areas, and at that time the list of endangered plant species was relatively short. Since 2018, monitoring using pyramid dead-in traps (AgBio Company) with aggregation pheromones (Tréce lures) has been carried out in agricultural and urban areas. The number of traps in the country increased from 20 to 47 in 2022 (Figure 1), and monitoring results showed a continuous spread throughout the country and an increase in the population. The list of endangered host plants has been significantly increased and includes both cultivated and ornamental plants [34]. Moderate to high-level damage has been reported in many cultivated plants since 2020, especially in hazelnuts, apples, cherries, peaches, and pears (Konjević, unpublished data). The protection of cultivated plants in Serbia relies almost entirely on chemical treatments, which have sometimes proven to be insufficiently effective.

Parasitoids of stink bugs, especially of exotic species, in Serbia were not monitored until 2022. In 2021, there were two accidental captures of three native parasitoid species, *Trissolcus basalis* (Wollaston), *Tr. belenus* (Walker), and *Telenomus* sp., which were morphologically identified at the Faculty of Biology, University “Al. I. Cuza” in Romania (Konjević and Popovici, unpublished data). Still, there was no evidence of the presence of exotic parasitoid species, or the autochthonous *A. bifasciatus*, to the eggs of *H. halys*.

Recent studies have estimated that in invaded areas, *Tr. japonicus* and *Tr. mitsukurii* are still spreading throughout much of the western Palearctic territory and have shown potential high habitat suitability in Serbia [28,29]. Therefore, the aim of this research was to monitor *H. halys* in urban and semi-urban environments and assess whether egg parasitoids are potentially present in such conditions.

**Figure 1 biology-13-00316-f001:**
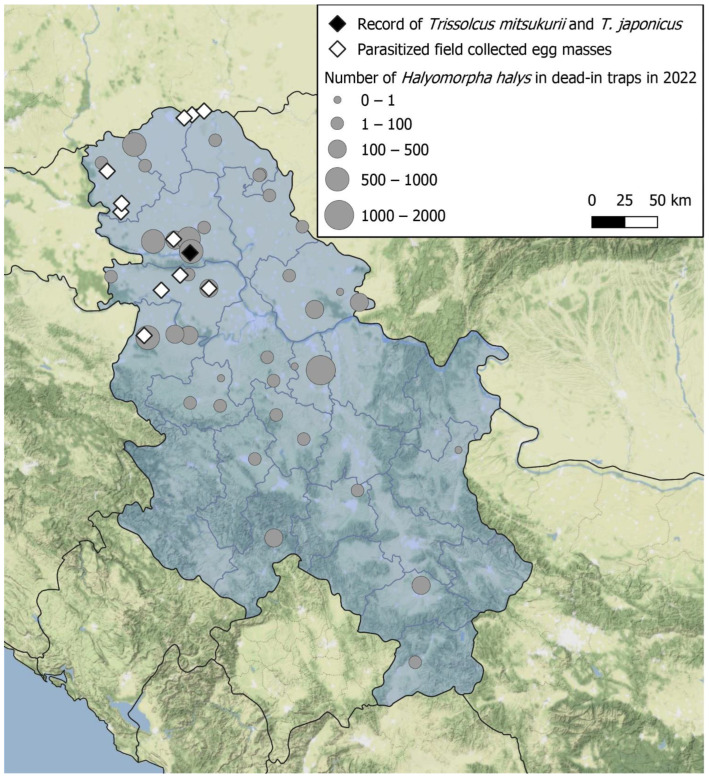
Sites with field-collected parasitized egg masses in 2022. Black rhombus: sites where *Tr. japonicus* and *Tr. mitsukurii* emerged from *H. halys* eggs. White rhombus: sites where only native parasitoid species were found. Gray circles: sites where dead-in traps were set in 2022 and the range of sampled *H. halys* specimens.

## 2. Materials and Methods

### 2.1. Field Collection and Laboratory Rearing of Stink Bug Egg Masses

The field survey was conducted from May to September 2022 at 27 different locations, mainly in the central and northern parts of Serbia, where the population of *H. halys* is regularly found in dead-in traps baited with aggregation pheromones. Firstly, the monitoring of egg masses started in the urban area of the second largest city in Serbia, Novi Sad, where three city parks were examined each week. Later, egg masses were sampled at the borders with Hungary, Croatia, and Romania in two visits during 2022, mainly urban and semi-urban environments. In addition, egg masses were occasionally collected in hazelnut, plum, and walnut orchards, and in a soybean field. All urban and semi-urban sites were characterized by a variety of plant species, but the search for eggs was focused on *Catalpa* sp., *Paulownia* sp., *Sambucus nigra* L., *Acer* sp., *Styphnolobium japonicum* (L.) Schott, and *Koelreuteria* sp., on which adults and nymphs of *H. halys* were observed. In many of these areas, there were no insecticide applications, except in orchards, where insecticide treatments were carried out regularly based on production needs.

Sampling in all areas was carried out in the morning hours, between 8 and 12 AM. The inspection time in each site was 1 h per person, looking at different parts of the same site. During the survey, woody and shrubby plant species were visually inspected by checking the leaves, especially the undersides where stink bugs usually lay their eggs. All field-collected stink bug eggs were brought to the laboratory and separated by sampling points and by insect species, comparing the morphology of the egg masses with images published on https://www.halyomorphahalys.com/wanzeneier-stink-bug-eggs.html (accessed on 7 February 2023) [35]. Eggs were then placed into petri dishes (10 or 15 cm in diameter) and stored under the room conditions for the emergence of either stink bug nymphs or adult parasitoids. All emerged parasitoids were stored in 96% ethanol at −4 °C, until their identification.

### 2.2. Species Identification

The parasitoid adults, previously stored in ethanol, were dried for morphological examination. A Wild M5 stereo microscope with up to 200× magnification was used for morphological diagnosis. Species of *Trissolcus* were determined using the characters and identification keys of Talamas et al. [36], Tortorici et al. [37], and Moraglio et al. [17]. Species of *Anastatus* were identified according to Peng et al. [38]. Species of *Telenomus* were determined using the keys of Kozlov and Kononova [39] and Johnson [40]. Images of the specimens were taken using a Canon 90D camera (Canon Inc., Tokyo, Japan) equipped with an extension tube and 10× and 20× LWD microscope lenses mounted on a macro-rail and illuminated with two speedlite flashes. The frames were merged with Zerene Stacker (PMax algorithm, Zerene Systems LLC, Richland, Washington, USA). All specimens used for morphological analysis were deposited in the Dipartimento di Scienze Agrarie, Forestali e Alimentari, University of Turin, Italy.

## 3. Results

A total of 568 naturally laid egg masses were collected, belonging to three pentatomid species: *H. halys*, *Nezara viridula* (L.), and an Asopinae (Hemiptera, Pentatomidae) (Table 1). Out of the 27 locations examined, parasitized egg masses were obtained from 12, reported in Table 1, while in the remaining locations either no eggs were recorded or only unparasitized eggs were found. Most of the eggs were collected on hazelnuts, in three orchards, while most of the parasitized eggs were sampled in the urban areas, on *St. japonicum* and *Sa. nigra*. Other host plants on which parasitized egg masses were collected are walnut, catalpa, plum, maple, paulownia, rose, and tomato. In total, parasitoid adults emerged successfully from 29 egg masses. Among the sampled eggs, there were eggs that were different in colour from the viable eggs, like dark yellow or brown, from which neither stink bug nymphs nor parasitoid adults emerged. Some eggs were already damaged by predators at the time of sampling.

Most of the parasitized eggs were found in areas where high numbers of *H. halys* specimens were monitored by dead-in traps, varying from 501 to 1000 per year/season (Figure 1).

*Anastatus bifasciatus* was the most abundant species emerging from the eggs of *H. halys* and *N. viridula* (Table 2). *Ooencyrtus* sp. emerged from the eggs of both *H. halys* and *N. viridula*, while *Telenomus turesi* Walker emerged only from the eggs of *H. halys* (Table 2). Based on morphological analyses, *Trissolcus* specimens were identified as *Tr. basalis*, *Tr. belenus*, *Tr. colemani* (Crawford), and *Tr. semistriatus* (Nees von Esenbeck), and also as the exotic *Tr. japonicus* and *Tr. mitsukurii*. These two exotic parasitoid species emerged at the site of Novi Sad from egg masses of *H. halys* (Figure 1).

*Trissolcus japonicus* emerged from four egg masses and *Tr. mitsukurii* from one egg mass together with the first. The two species perfectly matched with the description in Talamas et al. [36] and morphological diagnosis in Sabbatini Peverieri et al. [41], and they did not show morphological anomalies that would complicate confirmation of species identity based on morphology alone. *Anastatus bifasciatus* emerged from nine egg masses of *H. halys* and four egg masses of *N. viridula*. Multiparasitism was recorded in four egg masses of *H. halys* and three egg masses of *N. viridula* with *Ooencyrtus* sp., *Te. turesis*, *Tr. basalis*, and *Tr. japonicus*. Furthermore, *Ooencyrtus* sp. shared the same egg masses together with *Tr. japonicus*.

*Trissolcus japonicus* and *Tr. mitsukurii* belong to the *flavipes*-group and the *basalis*-group, respectively [36]. The two species can be distinguished from other species in their respective groups by the following characteristics:

*Trissolcus japonicus*—Vertex between lateral ocelli with uniform and robust hyperoccipital carina (hoc) (Figure 2B,E). Microsculpture finely colliculate along the line between the dorsal margin of the hyperoccipital carina and the posterior margin of the anterior ocellus (haol) (Figure 2E). At intersection with malar sulcus, orbital furrow (of) expanded with medial margin well-defined (Figure 2D). Clypeus with four setae (cs) (Figure 2C). Antenna with five clavomeres (A7–A11 with basiconic sensilla) with moderately large clava (Figure 2A). Mesoscutum with notauli (not) and without median mesoscutal carina (mmc) (Figure 2B,E). The finely colliculated sculpture on the mesoscutum is uniformly extensive, reaching the posterior margin between the notauli (Figure 2B,E). Mesopleuron with episternal foveae (eps) forming a continuous line of cells from postacetabular sulcus (ats) to mesopleural pit (mpp) (Figure 2D). Sublateral seta absent on each side of tergite 1 (Figure 2B,E). Laterotergite 1 without a line of setae along dorsal margin (Figure 2A,D). Tergite 2 with striae present throughout anterior half of tergite (Figure 2B).

*Trissolcus mitsukurii*—Vertex between lateral ocelli without hyperoccipital carina (Figure 3B,E). Clypeus with six setae (cs) (Figure 3C). Antenna with five clavomeres (A7–A11 with basiconic sensilla) with distinctly large clava (Figure 3A). Mesoscutum with notauli (not) (Figure 3B,E). Mesopleuron with two episternal foveae (eps) distant from postacetabular sulcus (ats) and mesopleural pit (mpp) (Figure 3D). Metapleuron without setae below metapleural sulcus (mtps) (Figure 3D). One or two sublateral seta (ss) present on each side of tergite 1 (Figure 3B,E). Laterotergite 1 with a line of setae along dorsal margin (slt1) (Figure 3C,D). Frons, from vertex to antennal scrobe, mesoscutum, and mesoscutellum coarsely rugose (Figure 3B,C,E).

## 4. Discussion

The parasitized egg masses collected at Novi Sad represent the first record of the exotic *Tr. japonicus* and *Tr. mitsukurii* in Serbia and indicate the presence of an established adventive population of both parasitoids. This is confirmed by the fact that there have been neither deliberate releases of these species for classical biological control nor any known laboratory colonies in the area. Models of the potential interaction between *H. halys* and these two parasitoid species hypothesized that *Tr. mitsukurii* and *Tr. japonicus* exhibit high and medium-high habitat suitability in Serbia, respectively [29], and the record of the adventive populations of these two parasitoids in the same area confirms what was predicted. For the purpose of evaluating a biological control program, the model proposed by the same authors argues that *Tr. mitsukurii* shows better suitability than *Tr. japonicus* in northern Serbia, but the habitat suitability of both scelionids overlaps in much of the country. By contrast, in the southern-most mountainous areas of the country, the probability of the suitability of both parasitoids is very low.

Since the two exotic parasitoids are well characterized and distinguishable from the western Palearctic species [36], only morphological identification was performed here. Further molecular-based investigations could assess the similarity of these adventive populations to the haplotypes distributed in the newly invaded and/or native areas and in this way trace the origin of the populations found in Serbia.

The means of arrival of *Tr. japonicus* and *Tr. mitsukurii* in Serbia is unknown, but we can suppose that their arrival is connected with the transport of goods and passengers. As the most likely means of arrival of *H. halys* was as a stowaway in the transport of goods and passengers [42,43,44], it is very likely that the parasitoid species also arrived in the same way [45]. Namely, as of 2020 there are several foreign construction companies located in Novi Sad and its surroundings, which have the majority of necessary equipment arriving from abroad. A few of them are located at sites close to the point where both exotic parasitoid species were found in a park. It should be emphasized that the traffic between the headquarters, or the companies’ place of business, and this park runs smoothly by the busy roads, leaving the possibility for their spread by vehicles. In contrast, in hazelnut orchards where the sampling was conducted in 2022 chemical treatments were carried out to control *H. halys*, which makes it difficult, although not impossible, to record and find potentially surviving parasitoid specimens. Also, hazelnut orchards are often located far from the main motorways; therefore, the spread of exotic parasitic species to those locations might need more time. These are the two main reasons why this survey was focused more on ornamental and not cultivated host plants in urban and semi urban areas.

The two species *Tr. japonicus* and *Tr. mitsukurii* can now be considered widespread throughout most European countries [18,27,30,31,41,46]. At present, in Serbia the two species were only found in one urban site and confirmed their preference for *H. halys* as the primary host. Since only a few specimens were found at one collection site, this may suggest that the spread of the adventive populations is in its early stage; therefore, the impact of the two parasitoids on the *H. halys* population cannot yet be evaluated. In subsequent years, the spread of both scelionids and their impact on the host population can be assessed by increasing the number of sites and expanding the collection season.

The Holarctic species *A. bifasciatus* is a generalist parasitoid [15,47], and its interspecific competition [48,49,50] makes it able to develop in different hosts and sharing a single egg mass together with other parasitoid species. Data obtained in the present study are consistent with those of other European studies and confirm the behaviour for this species. Very interesting are the data on other indigenous parasitoid species that emerged from *H. halys* eggs in a quantity similar to *A. bifasciatus*, particularly *Tr. basalis* and *Te. turesis*, which therefore deserve special attention to assess their possible adaptation and contribution to the control of the exotic pest. Due to the limited number of specimens of these native species found in the eggs of *H. halys*, it is difficult to predict if their presence has already affected or will potentially affect the decline of the alien and/or autochthonous stink bug species.

Given the economic importance of *H. halys* in Serbia, it will likely be beneficial to attract exotic parasitoids to agricultural fields and commercial fruit orchards, e.g., by increasing ecological infrastructures [51,52], and support them in a conservation biological control approach [53]. In this context, it may be warranted to further investigate the current status of the species in terms of spatial distribution and range of native hosts. Although laboratory studies were recently carried out [23,25,54], the interactions between populations of *Tr. japonicus* and *Tr. mitsukurii* in the field deserve close attention.

## 5. Conclusions

The brown marmorated stink bug, *H. halys*, is a serious threat to plant protection worldwide. Control strategies for this pest in Europe and America rely mainly on repeated application of chemical pesticides. From the perspective of an integrated control strategy with low pesticide use, attention has turned to natural enemies, such as egg parasitoids, which can be a valid strategy. To ensure the effectiveness of biological, a good knowledge of the pest and associated BCAs, including their biology, distribution, and diversity, is needed.

This study reports for the first time the presence of *Tr. japonicus* and *Tr. mitsukurii* in Serbia, thus providing additional valuable information on the geographic distribution of the two exotic parasitoids. Several countries are evaluating the introduction of these two parasitoids in classical biological control programs, and in some countries the release of either species is already underway. Knowing the spontaneous spread of both species is of paramount importance and can help better target and rationalize the control means, and guide scientists in implementing successful biological control strategies.

## Figures and Tables

**Figure 2 biology-13-00316-f002:**
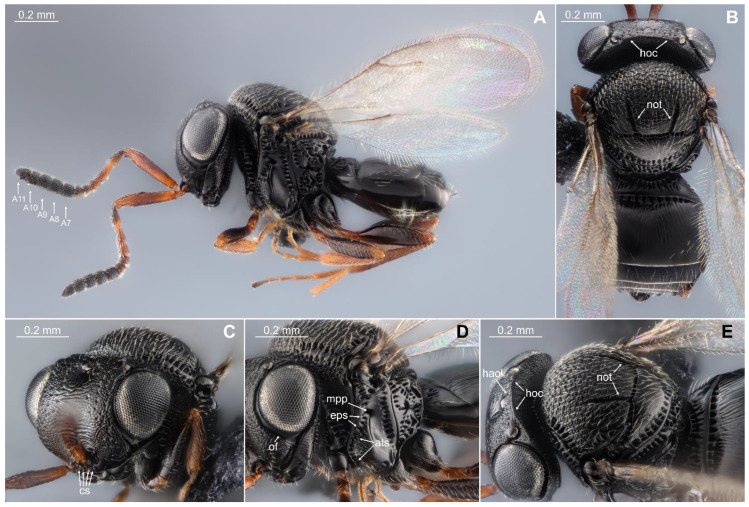
Female of *Tr. japonicus*. (**A**) head, mesosoma, metasoma, lateral view; (**B**) head, mesosoma, metasoma, dorsal view; (**C**) head, anterolateral view; (**D**) head, mesosoma, lateral view; (**E**) head, mesosoma, metasoma, dorsolateral view. A7–A11 = 5 clavomers; hoc = hyperoccipital carina; not = notauli; cs = clypeal setae; of = orbital furrow; mpp = mesopleural pit; eps = episternal foveae; ats = postacetabular sulcus; haol = line between the dorsal margin of hyperoccipital carina and posterior margin of anterior ocellus.

**Figure 3 biology-13-00316-f003:**
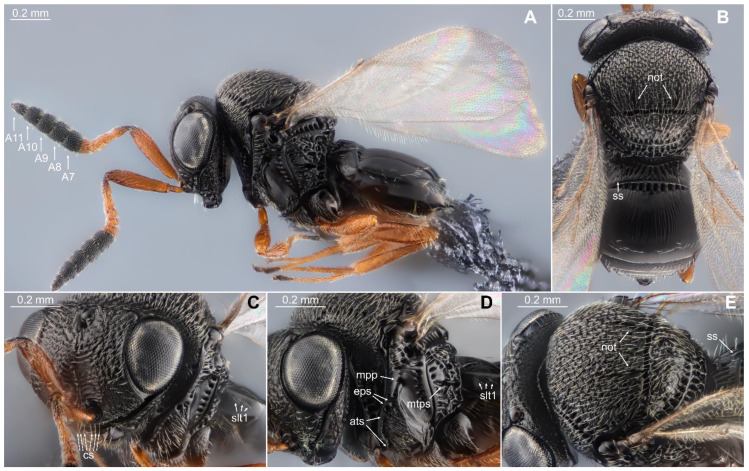
Female of *Tr. mitsukurii*. (**A**) head, mesosoma, metasoma, lateral view; (**B**) head, mesosoma, metasoma, dorsal view; (**C**) head, mesosoma, anterolateral view; (**D**) head, mesosoma, lateral view; (**E**) head, mesosoma, metasoma, dorsolateral view. A7–A11 = 5 clavomers; not = notauli; ss = sublateral seta; cs = clypeal setae; slt1 = setation of laterotergite 1; mpp = mesopleural pit; eps = episternal foveae; ats = postacetabular sulcus.

**Table 1 biology-13-00316-t001:** Sites in Serbia, where parasitized stink bug egg masses were collected in 2022. Hh, *Halyomorpha halys*; Nv, *Nezara viridula*; As, Asopinae. The asterisks (*) indicate cases of multiparasitisms.

ID	Site	Habitat	Coordinates	Altitude (m asl)	Total Number of Egg Masses	Parasitized Egg Masses
1	Backi Petrovac	Hazelnut orchard	45°19′38″ N 19°41′24″ E	80	1	Hh (1)
2	Backi Vinogradi	Semi-urban rural area, border with Hungary	46°7′26″ N 19°51′34″ E	95	2	Hh (2)
3	Horgos	Semi-urban rural area, border with Hungary	46°8′50″ N 19°58′11″ E	82	5	Hh (1)
4	Karavukovo	Rural area, border with Croatia	45°30′10″ N 19°12′22″ E	82	2	Hh (1 *)
5	Lacarak	Semi-urban rural area	44°59′49″ N 19°34′37″ E	81	3	Nv (2)
6	Ljukovo	Hazelnut orchard	45°0′32″ N 20°0′52″ E	101	306	Hh (1)
7	Mala Remeta	Agricultural area, orchards	45°5′40″ N 19°45′1″ E	205	10	Hh (1)
8	Novi Sad	Urban park	45°14′23″ N 19°50′36″ E	100	71	Hh (6 *), Nv (3 *), As (1)
9	Palic	Semi-urban rural area	46°6′14″ N 19°47′14″ E	98	1	Hh (1)
10	Ribari	Hazelnut orchard	44°42′9″ N 19°25′10″ E	93	150	Hh (3)
11	Sombor	Walnut orchard	45°45′53″ N 19°4′48″ E	84	12	Hh (2 *), Nv (3 *)
12	Srpski Miletic	Semi-urban rural area	45°33′27″ N 19°12′46″ E	83	5	Nv (1)

**Table 2 biology-13-00316-t002:** Total number of individuals of *A. bifasciatus* (Ab), *Ooencyrtus* sp. (Oo), *Te. turesis* (Tt), *Tr. basalis* (Tba), *Tr. belenus* (Tbe), *Tr. colemani* (Tc), *Tr. japonicus* (Tj), *Tr. mitsukurii* (Tm), *Tr. semistriatus* (Ts) emerged from *H. halys*, *N. viridula*, and Asopinae egg masses.

Species	No. Egg Masses	No. and Species of Parasitoids Emerged in Laboratory
*Halyomorpha halys*	19	49 Tba, 32 Ab, 31 Tj, 19 Tt, 10 Oo, 3 Tc, 2 Tm
*Nezara viridula*	9	62 Ab, 37 Tba, 26 Ts, 1 Oo
Asopinae	1	4 Tbe

## Data Availability

The raw data supporting the conclusions of this article will be made available by the authors on request.

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
