# Peer review of "The First Records of Trissolcus japonicus (Ashmead) and Trissolcus mitsukurii (Ashmead) (Hymenoptera, Scelionidae), Alien Egg Parasitoids of Halyomorpha halys (Stål) (Hemiptera, Pentatomidae) in Serbia"

_biology, 2024, doi:10.3390/biology13050316_

Round 1

Reviewer 1 Report

Comments and Suggestions for Authors

This is a rather simple study in  which you surveyed egg masses of a newly introduced stink bug and others for parasitoids. Finding  number of local  and two exotic parasitoids is valuable information towards the use of biological control against the pest. The taxonomic information and experience gained in identfying the parasitoids are also valuable as these important organisms are notoriously difficult to identify. It is commendable that you have given them to a museum.

I have  a few small points to consider:

Abstract: Scientific name of bug should be in italics as for the parasitoids. You should indicate that you recorded (found) the local parasitoid species.

Line 56. Write:  ... and then the parasitoids spread to France

60. Rather write: At that time the list......

100.  Species of what? Stink bugs? (Could be plants). The link only shows one species.

103. The use of "'parasitoids'' rather that "wasps" is preferable.

151-154. This is a rather clumsy sentence. Break it up after viridula, In the rest deal with multiparasitism. You cannot really record an egg mass as "multiparatisism". Write:  multiparatisiism was recorded in four egg masses..... with.....

157. Could you perhaps give  a reference for these groupings.

209. ..record of the exotic...

250. Could you elaborate  somewhat on support and attraction? Although you give a reference what do you have in mind? There is also the possibility of mass rearing of .parasitoids to be considered.

Comments on the Quality of English Language

The English language is good.

Author Response

Thank you very much for taking the time to review this manuscript. Please find the detailed responses below and the corresponding corrections in track changes in the re-submitted files. We also made other small changes to correct some typos that you can see in the track-changed version.

3. Point-by-point response to Comments and Suggestions for Authors

Comment 1: Abstract: Scientific name of bug should be in italics as for the parasitoids.

Response 1: Thank you for pointing this out. We wrote it in italics (line 19).

Comment 1 bis: You should indicate that you recorded (found) the local parasitoid species.

Response 1 bis: We agree with reviewer comment. Therefore, we changed lines 28–29 “European species” in “local species”.

Comment 2: Line 56. Write:  ... and then the parasitoids spread to France

Response 2: We have, accordingly, wrote: “….and then the parasitoid spread to France”.

Comment 3: 60. Rather write: At that time the list......

Response 3: We have, accordingly, wrote: “… at that time the list of …”.

Comment 4: 100. Species of what? Stink bugs? (Could be plants). The link only shows one species.

Response 4: Thank you for pointing this out. We modified the sentence il lines 99–100 and updated the text with the correct link:

“… and by insect species comparing the morphology of the egg masses with images published on https://www.halyomorphahalys.com/wanzeneier-stink-bug-eggs.html ...”.

Comment 5: 103. The use of "'parasitoids'' rather that "wasps" is preferable.

Response 5: We agree with this comment. Therefore, we used “parasitoids” in line 104 as well in the next sentence (same line). We wrote: “nymphs or adult parasitoids. All emerged parasitoids were stored in 96% ethanol at -4°C, until their identification”.

Comment 6: 151-154. This is a rather clumsy sentence. Break it up after viridula, In the rest deal with multiparasitism. You cannot really record an egg mass as "multiparatisism". Write:  multiparatisiism was recorded in four egg masses ..... with.....

Response 6: We agree, therefore we modified the sentence (lines 152–154):

“… viridula. Multiparasitism was recorded in four egg masses of H. halys and three egg masses of N. viridula with Ooencyrtus …”.

Comment 7: 157. Could you perhaps give a reference for these groupings.

Response 7: We have, accordingly, done. We cited the reference “[28]” in line 157.

The reference is “28.    Talamas, E.J.; Buffington, M.L.; Hoelmer, K.A. Revision of Palearctic Trissolcus Ashmead (Hymenoptera, Scelionidae). Journal of Hymenoptera Research 2017, 56, 3, doi:10.3897/JHR.56.10158”.

Comment 8: 209. ..record of the exotic...

Response 8: We agree, and corrected:

“…the first record of the exotic Tr. japonicus ”.

Comment 9: 250. Could you elaborate somewhat on support and attraction? Although you give a reference what do you have in mind? There is also the possibility of mass rearing of .parasitoids to be considered.

Response 9: We have, accordingly, modified the sentence to make it clearer:

“Given the economic importance of this invasive stink bug, it will likely be beneficial to at-tract these parasitic wasps to agricultural fields and commercial fruit orchards, e.g., by in-creasing ecological infrastructures [45,46], and support them in a conservation biological control approach [47].”

Therefore, we modified the reference list adding two new references “45,46”. The new list is:

45.   Olsen, D.M.; Wäckers, F.L. Management of field margins to maximize multiple ecological services. Journal of Applied Entomology 2007, 44, 13–21, doi:10.1111/j.1365-2664.2006.01241.x.

46.   Winkler, K.; Wäckers, F.L.; Pinto, D.M. Nectar‐providing plants enhance the energetic state of herbivores as well as their parasitoids under field conditions. Ecological Entomology 2009, 34, 221–227, doi:10.1111/j.1365-2311.2008.01059.x

47.   Abram, P.K.; Mills, N.J.; Beers, E.H. Review: Classical Biological Control of Invasive Stink Bugs with Egg Parasitoids – What Does Success Look Like? Pest Management Science 2020, 76, 1980–1992, doi:10.1002/PS.5813.

48.   Scala, M.; Fouani, J.M.; Zapponi, L.; Mazzoni, V.; Wells, K.E.; Biondi, A.; Baser, N.; Verrastro, V.; Anfora, G. Attraction of Egg Parasitoids Trissolcus mitsukurii and Trissolcus japonicus to the Chemical Cues of Halyomorpha halys and Nezara viridula. Insects 2022, 13, doi:10.3390/INSECTS13050439

Reviewer 2 Report

Comments and Suggestions for Authors

The article described Serbia's recent encounter with the Brown Marmorated Stink Bug, Halyomorpha halys, which was first documented in October 2015. This invasive species poses a significant threat to plant life, particularly during winter when it congregates in urban areas. Despite the surge in population causing substantial economic harm to crops, Serbia lacks natural predators for the stink bug. Consequently, research in 2022 focused on identifying egg parasitoids, revealing two previously unrecorded Hymenoptera species: Trissolcus japonicus and Tr. mitsukurii. Additionally, the study identified seven native European egg parasitoid species, marking the first report of nine such species parasitizing BMSB and related species in Serbia.

In my opinion the manuscript is interesting for biologists as well as for general readers.

However, I have question about identification of species:  Is identification based solely on morphological characteristics sufficiently accurate? Did the authors consider molecular identification for parasitoid species?

Author Response

Thank you very much for taking the time to review this manuscript. Please find the detailed responses below and the corresponding revisions/corrections highlighted/in track changes in the re-submitted files. We also made other small changes to correct some typos that you can see in the track-changed version.

Comment 1: However, I have question about identification of species:  Is identification based solely on morphological characteristics sufficiently accurate? Did the authors consider molecular identification for parasitoid species?

Response 1: Thank you for pointing this out. We chose to make the identification only on a morphological basis because the two species are now as well characterized as any other Palearctic species. Certainly, the analysis on a molecular basis could be useful to know the affinity of the specimens of the adventive population in Serbia with the populations from the native and newly spread areas. For this reason, we are enhancing research to know the real distribution of the two parasitoids in the area and whether more than one strain of the same species is present.

Therefore, we added a sentence in the “Discussion” (lines 222–225): “Since the two exotic parasitoids are well characterized and distinguishable from the western Palearctic species [28], only morphological identification was carried out. Further molecular-based investigations could assess the similarity of these adventive populations to the haplotypes distributed in the newly invaded and/or native areas.”
